# Disordered Social Media Use during COVID-19 Predicts Perceived Stress and Depression through Indirect Effects via Fear of COVID-19

**DOI:** 10.3390/bs13090698

**Published:** 2023-08-22

**Authors:** Gabriel Tillman, Evita March, Andrew P. Lavender, Taylor A. Braund, Christopher Mesagno

**Affiliations:** 1Institute of Health and Wellbeing, Federation University Australia, Ballarat, VIC 3350, Australia; e.march@federation.edu.au (E.M.); a.lavender@federation.edu.au (A.P.L.); chris.mesagno@vu.edu.au (C.M.); 2Black Dog Institute, University of New South Wales, Sydney, NSW 2031, Australia; t.braund@blackdog.org.au; 3Faculty of Medicine and Health, University of New South Wales, Sydney, NSW 2052, Australia; 4Institute for Health & Sport, Victoria University, Melbourne, VIC 3011, Australia

**Keywords:** fear of COVID-19, disordered social media use, depression, stress

## Abstract

The 2019 novel coronavirus disease (COVID-19) is a global threat that can have an adverse effect on an individuals’ physical and mental health. Here, we investigate if disordered social media use predicts user stress and depression symptoms indirectly via fear of COVID-19. A total of 359 (timepoint 1 = 171, timepoint 2 = 188) participants were recruited via social media and snowball sampling. They completed an online survey that measured disordered social media use, fear of COVID-19, perceived stress, and depression symptomatology at two cross-sectional timepoints. We found that disordered social media use predicts depression indirectly through fear of COVID-19 at both timepoints. We also found that disordered social media use predicts perceived stress indirectly through fear of COVID-19, but only at timepoint 1. Taken together with previous research, our findings indicate that disordered social media use may lead to increased fear of COVID-19, which in turn may lead to poorer psychological wellbeing outcomes. Overall, there is evidence that the impact of the COVID-19 pandemic is affecting the physical, psychological, and emotional health of individuals worldwide. Moreover, this impact may be exacerbated by disordered use of social media.

## 1. Introduction

The COVID-19 pandemic has significantly impacted the psychological and physical wellbeing of billions across the world [1], with a reported 598,349,130 cases and 6,464,663 deaths as of August 2022 [2]. This global issue has created a need for further study of COVID-19 and its impact on health and overall mental health, and whether any forms of interventions can assist in minimising or alleviating such impact. In the current study, we explore for the first time how disordered social media use relates to fear of COVID-19, and how in turn this may relate to psychological wellbeing.

### 1.1. COVID-19 and Depression and Stress

Two recent studies from China reported high levels of psychological distress during the initial stages of the pandemic [3]. In Australia, depression and stress scores during the pandemic are slightly elevated compared to normative data for Australian adults [4]. Some key factors associated with COVID-19 psychological distress include having pre-existing mental health conditions, increased smoking and alcohol drinking behaviour, high levels of fear, and being female [5]. In other countries, such as Italy, depression, anxiety, and stress scores were even higher than the study by Stanton et al. [5,6]. In a UK study by McPherson et al. [7], one tenth of nearly 2000 participants presented consistently high mental health symptoms over a 12-week period during the pandemic, with one twentieth of the sample also displaying clinically significant symptomology over the 12-week period. McPherson et al. [7] found that previous mental health issues were a key predictor for the participants psychological wellbeing. Overall, we are seeing psychological impacts of COVID-19 across the world.

Psychological factors that were products of the COVID 19 pandemic were also identified as key causes of suicide during COVID-19 [8,9,10]. Such factors included fear of COVID-19 infection, financial crisis, loneliness, pressure to be quarantined, being COVID-19 positive, COVID-19 work-related stress, unable to return home due to lockdown, and unavailability of alcohol [8]. Alarmingly, recent modelling studies suggest that future suicide rates will increase by up to 145% globally because of the pandemic [9,10]. Overall, these studies demonstrate that COVID-19 and related factors causes distress and other negative psychological outcomes, and there is a general trend, internationally, of decreased psychological wellbeing during COVID-19.

### 1.2. Disordered Social Media Use and Depression and Stress

Disordered social media use is defined as time-consuming use of social media that leads to symptoms such as tolerance, withdrawal, salience, mood modification, relapse, conflict, and negative impact on one’s quality of life [11]. People who demonstrate disordered social media use spend an excessive amount of time on social media, often to the detriment of their personal relationships, work or school performance, and other important aspects of life. They experience withdrawal symptoms when attempting to stop and then the continued obsessive use of social media increases anxiety, apathy, depressed mood, and a sense of isolation from social reality [12].

There is a growing body of research that has found an association between disordered social media use and depression and stress, among other negative mental health outcomes. Higher disordered social media use has consistently been found to have a positive association with higher depression and stress symptoms [13,14,15]. One possible explanation for the link between disordered social media use and depression and stress is that disordered social media use can lead to social comparison and feelings of inadequacy and isolation, which in turn can lead to negative mental health outcomes [11]. Specifically, constant exposure to idealized images and carefully curated content on social media leads to unhealthy social comparisons and negatively impacts one’s self-esteem. Another potential mechanism is that disordered social media use is the impact on sleep, which is known to be important for mental health, as excessive use of social media, especially before bedtime, can disrupt sleep patterns and quality, leading to sleep deprivation [16]. Here, we explore the possibility that disordered social media use may be related to higher levels of fear of COVID-19, which may be related to being over exposed to COVID-19 related news.

### 1.3. Fear of COVID-19

Fear of COVID-19 can be defined as an overwhelming and irrational fear of contracting the COVID-19 virus or of its potential consequences, which causes distress and impairment in daily life [17]. Fear of COVID-19 has been associated with increased depression and stress. For instance, a study by Humphrey et al. [18] found that an increase in fear of COVID-19 was associated with increased depression and stress scores. In terms of social media, a study by Khan, Ismail, and Gul [19] suggests that Facebook was the most popular source of obtaining any form of news, including COVID-19 related news, and that frequency of Facebook use was positively related to fear of COVID-19. Moreover, problematic social media use during COVID-19 is related to both fear of COVID-19 and depression [20]. Together, fear of COVID-19 appears to be related to depression and stress, and one key source of fear of COVID-19 could be attributed to disordered use of social media during COVID-19.

### 1.4. Stressor–Strain–Outcome Model

To understand the associations between disordered social media use, fear of COVID-19, and psychological distress, we apply the stressor–strain–outcome model (SSO) [21], a framework that has previously been used to understand social media use and psychological outcomes [22,23,24]. In this model, stressors are situations, stimuli, or events that can elicit a stress response in individuals. They can be external, such as major life events, work demands, or social conflicts, or internal, such as personal worries or health concerns. These stressors lead to strains, which refers to the psychological, emotional, and physiological responses experienced by an individual when faced with stressors [25]. The outcomes of the stressor–strain–outcome model represent the consequences or results of the stress and strain experienced by an individual. The model acknowledges that outcomes can, in turn, become stressors themselves, leading to a feedback loop. For example, strain resulting from work-related stress may negatively impact job performance, which then contributes to further stress [21]. Statistically speaking, the strain mediates the relationship the stressor has on the outcome. In our example, disordered social media use is the stressor, and increased fear of COVID-19 is the strain. This in turn leads to the outcome of increased depression and stress (see Figure 1).

In summary, COVID-19 related news accessed via social media has been shown to be positively related to depression and stress. This is because disordered social media use may be related to higher levels of fear of COVID-19, where fear of COVID-19 itself is associated with depression and stress. Given that we are seeing psychological impacts of COVID-19 across the world, it is important to explore the combination of disordered social media use, fear of COVID-19, and depression and stress. The aim of the current study was to apply the SSO model to explore how disordered social media use during COVID-19 can predict stress and depression (i.e., outcomes) via the strain of increased fear of COVID-19. In the current study, we surveyed participants on their disordered social media use, fear of COVID-19, depression, and stress at two cross-sectional timepoints during the COVID-19 lockdowns in Australia. We hypothesised that disordered social media use will positively predict both depression and stress and that there will be significant indirect pathways via fear of COVID-19.

## 2. Materials and Methods

### 2.1. Participants

Participants were recruited globally at two timepoints. At timepoint 1 (July 2020), 171 participants (86 male, 85 female) between the ages of 19 and 80 years (*M_age_* = 39.2, *SD_age_* = 13.4) participated. At timepoint 2 (December 2020), 188 participants (101 male, 85 female, 2 transgender) between the ages of 18 and 62 years (*M_age_* = 33.5, *SD_age_* = 9.47) participated. Inclusion criteria were that participants were aged 18 years and over and were fluent in English.

### 2.2. Measures

#### 2.2.1. Demographics

The demographic questionnaire included information about the participants’ gender and age. We also assessed frequency of daily social media use (Van Den Eijnden et al., 2016) [26] pre and during the COVID-19 pandemic.

#### 2.2.2. The Social Media Disorder Scale

We modified the 27-item Social Media Disorder Scale (SMDS) [26] to assess disordered use of social media during the COVID-19 pandemic. Specifically, we modified the original lure of “During the past year, have you…” to read, “During the COVID-19 pandemic, have you…”. The measure comprises 9 subscales: Preoccupation, Tolerance, Withdrawal, Persistence, Escape, Problems, Deception, Displacement, and Conflict. Example items include “often sat waiting until something happens on social media again?” (Preoccupation); “felt the need to use social media more and more often?” (Tolerance). Participants responded Yes (scored 1) or No (scored 0) to each statement. Total scores were calculated by summing responses with higher scores indicating higher levels of disordered social media use. Internal consistency was excellent (α = 0.96).

#### 2.2.3. Fear of COVID-19 Scale

The Fear of COVID-19 Scale (FCV-19S) [27] was included to assess fear of COVID-19. The FCV-19S comprises seven items (e.g., “I am most afraid of coronavirus-19” and “I cannot sleep because I’m worrying about getting coronavirus-19”; α = 0.92). Participants responded on a 5-point Likert scale ranging from 1 (strongly disagree) to 5 (strongly agree). Total scores are computed by summing the responses with higher scores indicating greater fear of Coronavirus. The internal reliability for this sample was adequate (α = 0.92).

#### 2.2.4. Perceived Stress Scale

The Perceived Stress Scale (PSS-10) [28] was included to measure an individual’s perceived stress over the past month. The PSS-10 includes 10 items (e.g., “In the last month, how often have you felt that you were unable to control the important things in your life?” and “In the last month, how often have you found that you could not cope with all the things you had to do?”; (α = 0.78). Participants respond to items on a 5-point Likert scale ranging from 0 (Never) to 4 (very often). After reverse scoring four items, total scores are calculated by summing responses, with higher scores indicating higher perceived stress. The internal reliability in this sample was adequate (α = 0.78).

#### 2.2.5. Centre for Epidemiologic Studies Depression Scale

The Centre for Epidemiologic Studies Depression Scale (CES-D) [29] was included to assess participants’ depressive symptoms in the past week. The measure comprises 20 items (e.g., “example item”; α = 0.93) and participants respond to items using a 4-point rating scale ranging from 0 (rarely or none of the time) to 3 (most or all of the time). Responses are summed for total scores with higher scores indicating more depressive symptoms. Cronbach’s alpha for the current sample was 0.93.

### 2.3. Procedures

Following the [BLIND FOR REVIEW] Human Research Ethics Committee approval, participants were recruited via a convenience sample and a snowballing technique on social media (e.g., Facebook, Twitter, and Instagram posts). The advertisements posted on social media included a weblink to the online questionnaire hosted by Qualtrics. Social media contacts were encouraged to forward the social media posts onto possible participants. Upon arriving at the questionnaire, participants were provided with an overview of the study and informed that by commencing the questionnaire they were providing their informed consent to participate. As an incentive, participants were offered a chance to win one (of two) AU$50 egift vouchers, which were randomly drawn from the pool of fully completed questionnaires once the current data collection period completed. Participants were informed that the questionnaire would take ~one hour to complete and upon completing the questionnaire were debriefed and thanked. The online data collection period remained open for one month from May to June 2020 for timepoint 1 and September to December 2020 for timepoint 2. A reminder post was sent out on social media to followers once a week.

### 2.4. Data Analysis

Analyses were conducted using the software R [30]. Before conducting our main mediation analyses, we used 4 independent samples *t*-tests to investigate differences between SMDS, FCV19S, PSS, and CESD. For both timepoints, we ran two mediation models using the “pathj” module [31] in the Jamovi package (The Jamovi Project, 2021) in R. The mediation analyses explored the possible indirect relationships between disordered social media use and negative emotional states (i.e., depression symptomatology and perceived stress) via fear of COVID-19. In all mediation analyses, age was included as a covariate to control for any influence of age in the model. To test for significant coefficients in the mediation models we used bias adjusted bootstrap confidence intervals with 5000 repetitions.

## 3. Results

Before conducting our mediation analyses, we first inspected if there were any changes in our key variables across the cross-sectional timepoints. These analyses were conducted after the data cleaning process for each timepoint was conducted, which are described below in their respective sections. Disordered social media use was significantly lower at timepoint 2 (*M* = 7.5, *SD* = 5.82) compared to timepoint 1 (*M* = 10.5, *SD* = 8.21), *t*(357) = 4.04, *p* < 0.001, *d =* 0.43, 95% CI [0.21, 0.64]. Fear of COVID-19 was significantly lower at timepoint 2 (*M* = 17.6, *SD* = 6.19) compared to timepoint 1 (*M* = 21.2, *SD* = 7.33), *t*(357) = 5.13, *p* < 0.001, *d =* 0.54, 95% CI [0.33, 0.76]. Perceived stress was significantly higher at timepoint 2 (*M* = 29.6, *SD* = 6.17) compared to timepoint 1 (*M* = 19.6, *SD* = 6.27), *t*(357) = −15.26, *p* < 0.001, *d =* −1.61, 95% CI [−1.88, −1.34]. There was no significant difference between depression scores at timepoints 1 and 2, *t*(357) = 1.85, *p* = 0.065, *d =* 0.20, 95% CI [−0.01, 0.40].

### 3.1. Timepoint One

#### 3.1.1. Data Screening

Before any analyses were conducted, we determined that there were ~4% of the data missing in total. Nine participants had more than 40% of their data missing. After removing these nine participants, the total amount of data missing was ~0.5%. However, some variables had up to ~3% of their data missing and some participants had ~9% missing. To deal with missing data we generated multivariate imputations by chained equations using the “Mice” package in R [32]. We ran a single imputation using the predictive mean matching method with 20 iterations. All imputed data sets were inspected via density plots to ensure the imputation process produced plausible values.

#### 3.1.2. Descriptive Statistics

Means, SDs, and bivariate correlations for all variables of interest are presented in Table 1. All bivariate correlations were significant except for the relationship between age and FCV19S.

#### 3.1.3. Mediation Analysis: Depression

Disordered social media use predicted depression while controlling for the fear of COVID-19, β = 0.51, *b* = 0.85, 95% CI [0.61, 1.10]. Disordered social media use predicted the fear of COVID-19, β = 0.68, *b* = 0.62, 95% CI [0.52, 0.72]. The fear of COVID-19 predicted depression while controlling for disordered social media use, β = 0.26, *b* = 0.48, 95% CI [0.21, 0.75]. Finally, there was an indirect effect in the model, β = 0.18, *b* = 0.30, 95% CI [0.13, 0.47].

#### 3.1.4. Mediation Analysis: Perceived Stress

Disordered social media use predicted perceived stress while controlling for the fear of COVID-19, β = 0.36, *b* = 0.26, 95% CI [0.14, 0.38]. The fear of COVID-19 predicted perceived stress while controlling for disordered social media use, β = 0.22, *b* = 0.17, 95% CI [0.04, 0.31]. Finally, there was an indirect effect in the model, β = 0.15, *b* = 0.11, 95% CI [0.02, 0.19]. Mediation paths are visualised in Figure 2.

### 3.2. Timepoint Two

#### 3.2.1. Data Screening

Like timepoint 1, and before any analyses were conducted, we determined that there were ~4% of the data missing in total. Six participants had more than 40% of their data missing. After removing these six participants, the total amount of data missing was ~1%. However, some variables had up to ~21% of their data missing and some participants had ~17% missing. To deal with missing data we generated multivariate imputations by chained equations. We ran a single imputation using the predictive mean matching method with 20 iterations. All imputed data sets were inspected via density plots to ensure the imputation process produced plausible values.

#### 3.2.2. Descriptive Statistics

Means, SDs, and bivariate correlations for all variables of interest are presented in Table 2. All bivariate correlations were significant except for the relationship between age and SMDS, CESD, and PSS.

#### 3.2.3. Mediation Analysis: Depression

Disordered social media use predicted depression while controlling for the fear of COVID-19, β = 0.54, *b* = 1.19, 95% CI [0.92, 1.46]. Disordered social media use predicted the fear of COVID-19, β = 0.39, *b* = 0.42, 95% CI [0.27, 0.56]. The fear of COVID-19 predicted depression while controlling for disordered social media use, β = 0.13, *b* = 0.28, 95% CI [0.02, 0.53]. Finally, there was an indirect effect in the model, β = 0.05, *b* = 0.11, 95% CI [0.003, 0.23].

#### 3.2.4. Mediation Analysis: Perceived Stress

Disordered social media use predicted perceived stress while controlling for the fear of COVID-19, β = 0.41, *b* = 0.43, 95% CI [0.29, 0.58]. The fear of COVID-19 did not predict perceived stress while controlling for disordered social media use, β = 0.12, *b* = 0.13, 95% CI [−0.008, 0.26]. Finally, there was no indirect effect in the model, β = 0.05, *b* = 0.06, 95% CI [−0.006, 0.11]. Mediation paths are visualised in Figure 2.

## 4. Discussion

The current study explored the relationship between disordered social media use and negative emotional states (i.e., stress and depression) during COVID-19, and if these variables were related indirectly via the fear of COVID-19. We found that disordered social media use positively predicted depression scores indirectly through fear of COVID-19 at both timepoints in our repeated cross-sectional design. Disordered social media use positively predicted perceived stress at timepoint 1 only. We also found that disordered social media use and fear of COVID-19 decreased between timepoint 1 and timepoint 2, whereas perceived stress increased.

This research was guided by the stressor–strain–outcome model (SSO) [21] that has been used to explore social media use and outcomes in the past [22,23,24]. Under this model, the increased social media use (i.e., stressor) is related to perceived stress and depression symptomatology (i.e., outcomes; see Figure 1). Moreover, the relationship between these stressors and outcomes is mediated by the fear of COVID-19 (i.e., strain). We noted that the fear of COVID-19 is an important strain because reading or hearing about the severity and contagiousness of COVID-19 is known to be stress inducing [33]. This could mean that during the COVID-19 pandemic, the impact of social media use on our perceived stress and depression may in part be driven by increased fear of COVID-19. On the other hand, disordered social media use still showed a direct relationship with stress and depression scores as the total effect was only partially mediated by fear of COVID-19 (8–10% mediation of the total effect at both timepoints). Therefore, disordered social media use is indirectly related to our stress and depression symptoms via fear of COVID-19, but also in other ways not explored in this study. Moreover, our work is consistent with the finding that an increased fear of COVID-19 is positively related to depression [18,34]. Finally, the relationships discussed above were present at both timepoints despite disordered social media use and fear of COVID-19 decreasing between timepoint 1 and timepoint 2. However, this is because the relationship between variables does not depend on the mean score but rather the rank order of individuals. Suggesting the pattern of individual differences is present at both timepoints.

The strength of our study is that it integrates many empirical findings related to disordered social media use, fear of COVID-19, and psychological wellbeing within the stressor–strain–outcome model in a single repeated cross-sectional study. What previous results and our study support is that people who have problematic use of social media show higher rates of stress, anxiety, depression, and other negative mental health issues [12,14,35]. More specifically, under the SSO model, the use of social media can incidentally expose users to COVID-19 related news, which is known to negatively affect users [36,37]. The stressor of social media use relates to levels of fear of COVID-19, which may be derived from the news consumed on social media sites and this higher level of fear of COVID-19 is related to higher levels of depression and stress [18,19,37].

The results of this study were cross-sectional and provide an insight into how disordered social media use is related to psychological wellbeing at sequential cross-sectional timepoints during COVID-19. The study was limited as we aimed to follow up on participants that would be willing to take part at both timepoint 1 and timepoint 2, yet only 20 participants were present at both timepoints. Therefore, we were not able to model changes in the relationships reported longitudinally over time. The cross-sectional nature of our work limits us to providing evidence for an indirect pathway from disordered social media use through fear of COVID-19 to psychological wellbeing. Future research should employ a longitudinal design to better evaluate the extent to which fear of COVID-19 mediates the relationship between disordered social media use and depression and stress. It would also be important to further expand the causal models tested here to explore the Preoccupation, Tolerance, Withdrawal, Persistence, Escape, Problems, Deception, Displacement, and Conflict dimensions of disordered social media use.

Overall, there is evidence that the impact of the COVID-19 pandemic is affecting the physical, psychological, and emotional health of individuals worldwide. And we suggest that the psychological impact may be exacerbated by content on social media that promotes the fear of COVID-19 to users.

## Figures and Tables

**Figure 1 behavsci-13-00698-f001:**
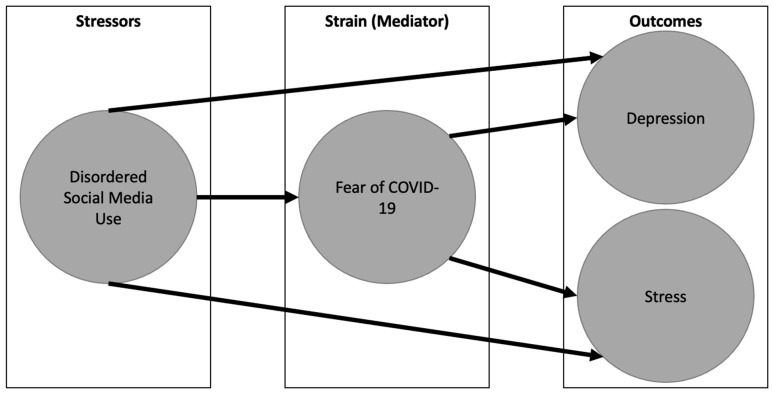
Graphical representation of the stressor–strain–outcome (SSO) model.

**Figure 2 behavsci-13-00698-f002:**
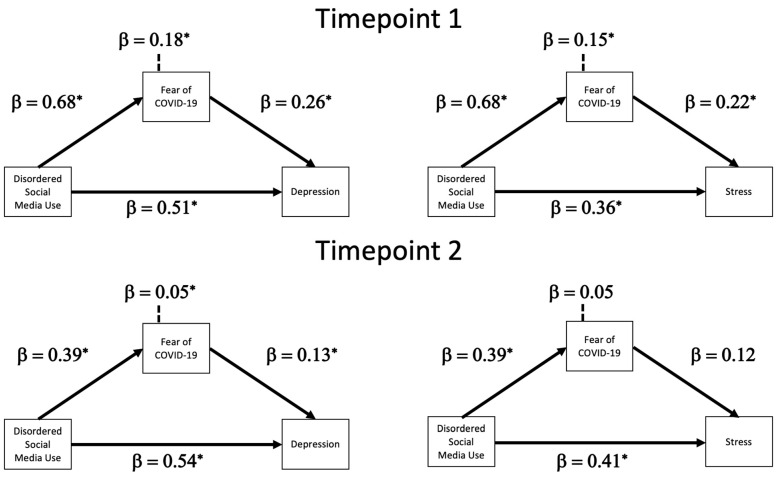
Graphical representation of all 4 mediation models. The standardised coefficients above the dashed line in each figure represent the indirect effect via the fear of COVID-19 pathway. The * indicates that the pathway was significant based on 95% CI crossing 0.

**Table 1 behavsci-13-00698-t001:** Descriptive statistics and correlations for study variables at timepoint 1.

Variable	*n*	*M*	*SD*	1	2	3	4	5
1. Age (Years)	171	39.2	13.4	—				
2. SMDS	171	10.5	8.21	−0.30 ***	—			
3. FCV19S	171	21.2	7.33	−0.04	0.68 ***	—		
4. CESD	171	22.5	13.5	−0.22 **	0.68 ***	0.60 ***	—	
5. PSS	171	19.6	6.27	−0.26 ***	0.51 ***	0.47 ***	0.70 ***	—

** *p* < 0.01 *** *p* < 0.001.

**Table 2 behavsci-13-00698-t002:** Descriptive statistics and correlations for study variables at timepoint 2.

Variable	*n*	*M*	*SD*	1	2	3	4	5
1. Age (Years)	188	33.5	9.68	—				
2. SMDS	188	7.5	5.82	−0.09	—			
3. FCV19S	188	17.6	6.19	0.14 *	0.39 ***	—		
4. CESD	188	20	12.7	−0.07	0.59 ***	0.35 ***	—	
5. PSS	188	29.6	6.17	−0.08	0.46 ***	0.29 ***	0.67 ***	—

* *p* < 0.05. *** *p* < 0.001.

## Data Availability

Not applicable.

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
