# Peer review of "Disordered Social Media Use during COVID-19 Predicts Perceived Stress and Depression through Indirect Effects via Fear of COVID-19"

_behavsci, 2023, doi:10.3390/bs13090698_

Round 1

Reviewer 1 Report

Dear Author

Thank you for submitting your paper to this outlet. I read your paper and gave my concern down here:

1. The title seems alright.

2. The abstract needs further attention to amend, meaning that the author might follow IMRaD concept.

3. The introduction section needs revision. Particularly, the author needs to show the background of the study and the research problem. The research problem must be novel and the gravity of it to solve. However, I don't observe the motivation/essence of the study.

4. In both times, T1 & T2, the sample size seems too small. The author must justify the adequacy of the sample size.

5. The author must demonstrate the sampling technique, sampling frame, and response rate.

6. Did the authors testify the reliability and validity of the data and scale?

7. The author might a section on Strengths/Contributions of the findings.

8. Moreover, the directions for future research must be added.

Wish you all the best.

Author Response

  1. The title seems alright.

The title has been updated to better reflect the paper's key findings.

“Disordered Social Media Use During COVID-19 Predicts Perceived Stress and Depression through Indirect Effects via Fear of COVID-19”

  1. The abstract needs further attention to amend, meaning that the author might follow IMRaD concept.

The abstract has been extended to include more information, in particular about the implications of the study. See Lines 20-24.

  1. The introduction section needs revision. Particularly, the author needs to show the background of the study and the research problem. The research problem must be novel and the gravity of it to solve. However, I don't observe the motivation/essence of the study.

Several paragraphs have been added to the introduction to better justify the study. They have been highlighted in yellow in the manuscript. See lines 61-64, 73-78, 82-84, 89-93, 98-107 and 114-118.

  1. In both times, T1 & T2, the sample size seems too small. The author must justify the adequacy of the sample size.

The discussion section now mentions the initial longitudinal design that aimed to collect two time points of the same participants. Given the attrition rates, we moved to two cross-sectional time points. The limitations of this are stated in the discussion. See lines 328-333.

  1. The author must demonstrate the sampling technique, sampling frame, and response rate.

We stated on lines 178 and 179… “participants were recruited via a convenience sample and a snowballing technique on social media (e.g., Facebook, Twitter, and Instagram posts)”. Moreover, we had no sampling frame or response rate because we did not send invitations out to specific populations, it was open to anyone on social media. See lines 184-185.

  1. Did the authors testify the reliability and validity of the data and scale?

Reliability estimates are now provided for each scale used in the study. See sections 3.1.3 and 3,1,4, lines 240-251.

  1. The author might a section on Strengths/Contributions of the findings. Moreover, the directions for future research must be added.

The final 2 paragraphs of the discussion section discuss the strengths, limitations, and contributions of the study. See lines 316-338.

Reviewer 2 Report

It is an interesting work, with a good introduction to the subject. I would just like to comment on a few issues: 

2.2. Measures

Quality data (through internal consistency) are only provided for the Social Media Disorder Scale (SMDS), but quality data are missing for all other scales used. It would be appropriate to provide the same data for all scales.

2.3. Procedures

Reading the procedure for obtaining the sample, the questions arise: Is the sample representative? What was the rejection rate?

Regarding figure 2, it is difficult to see the letters and specially the numbers  because (perhaps) of the low quality of the image, they are "out of focus".

4. Discussion

Lines 274-275. "Moreover, our work is consistent with the finding that an increased fear of COVID-19 is negatively related to depression [17,33]". I don't quite understand the meaning of this phrase. From the results shown, a greater fear of COVID is related to higher levels of depression, so wouldn't the relationship be positive?

Author Response

2.2. Measures

Quality data (through internal consistency) are only provided for the Social Media Disorder Scale (SMDS), but quality data are missing for all other scales used. It would be appropriate to provide the same data for all scales.

Reliability estimates are now provided for each scale used in the study. See sections 3.1.3 and 3,1,4, lines 240-251.

2.3. Procedures

Reading the procedure for obtaining the sample, the questions arise: Is the sample representative? What was the rejection rate?

We stated on lines 178 and 179… “participants were recruited via a convenience sample and a snowballing technique on social media (e.g., Facebook, Twitter, and Instagram posts)”. Moreover, we had no sampling frame or response rate because we did not send invitations out to specific populations, it was open to anyone on social media. See lines 184-185.

  1. Discussion

Lines 274-275. "Moreover, our work is consistent with the finding that an increased fear of COVID-19 is negatively related to depression [17,33]". I don't quite understand the meaning of this phrase. From the results shown, a greater fear of COVID is related to higher levels of depression, so wouldn't the relationship be positive?

Thank you for pointing this out, we have now fixed this in the manuscript:

“Moreover, our work is consistent with the finding that increased fear of COVID-19 is positively related to depression [17,33].” See lines 309-310.

Author Response

  1. Sample size and justification: The authors should provide a rationale for the chosen

sample size in the research. Explaining why the sample size was considered appropriate for addressing the research objectives is essential. Additionally, the study was conducted twice with different sample sizes and groups. In that case, the authors should clarify the reasons behind this decision, highlighting any potential differences or advantages gained from these variations.

The discussion section now mentions the initial longitudinal design that aimed to collect two time points of the same participants. Given the attrition rates we moved to two cross-sectional time points. The limitations of this are stated in the discussion. Please also see response to reviewer 1 about our sample. See lines 310-312.

  1. Age group selection: The authors should provide a clear justification for including a broad age range in their study, ranging from 19 to 80 years in Study 1 and 18 to 62 years in Study

The sampling criteria allowed participants above 18 to participate in the study, and therefore the age range differences were due to sampling and not due to any differences in inclusion criteria.

3. Given that younger individuals are often more technologically savvy, it would be beneficial to explain why a more comprehensive age range was chosen, especially when investigating the impact of disordered social media use. This justification should consider potential variations in social media use patterns across different age groups.

This is an interesting point regarding a potential covariate to the study. Unfortunately, we did not measure any technology capacity constructs in the study and would not be able to dive into this research question in the study. Moreover, our mediation included age as a covariate for all models to limit this potential confound.

4. Demographic details and key variables: It is vital to include demographic information and key variables in the form of tables. This will provide a comprehensive overview of the sample characteristics and facilitate a better understanding of the study's findings. By presenting this information in a tabular format, readers can quickly assess the distribution of variables and their potential associations.

Table 1 and 2 contain summaries of all the key variables considered in this study at time point 1 and 2, respectively. See lines 237 and 266.

5. Clarify software used for mediational analysis: In lines 169-170, there appears to be confusion regarding the software used for mediational analysis. The authors should explicitly state whether they used Jamovi or R software. Providing this clarification will ensure the accuracy and reproducibility of the study.

Jamovi has a package that can be implemented in R, and so although Jamovi is a stand alone software we used R with the Jamovi package. This has been clarified in text. Lines 202-203.

6. Consistency in reporting results: The authors should consistently report standard errors (SE), p values, and clearly mention the un/standardized coefficient in the mediation analysis results. Additionally, it is important to consistently use a leading zero (or not using zero) before decimal values throughout the paper to ensure clarity and precision.

We have gone through and corrected all statistics presentations, including leading zeros for each coefficient in text and differentiating standardised and unstandardised coefficients. Figure 2 now mentions that the coefficients in the figure are standardised. Moreover, confidence intervals are presented for each inferential test but not p-values given our bootstrap methodology.

7. Improve clarity of figures: The figures presented should clearly indicate which paths are significant or insignificant. Furthermore, in Figure 2, for Time point 2, the authors appear to show both figures as indirectly affected by Fear of COVID-19, which contradicts the results section. The authors should carefully review the figures to ensure accuracy and consistency with the reported results.

 Figure 2 now has * to indicate significant paths. Leading zeros, text size, and overall clarity of the figures has been improved. See line 282.

8. Justify mediation analysis in cross-sectional data: The authors should justify using mediation analysis in cross-sectional data at a single time point. It is important to explain why this approach was chosen and discuss any potential limitations of this analysis method.

We have changed the terminology throughout to use the words indirect effect to indicate that these path models are not testing full mediation, which would require longitudinal data.

9. Discuss contradictory results and variations across time points: The authors should address the baffling results observed in the second time point, where disordered social media use and fear of COVID-19 decrease, yet depression is still predicted. The discussion section should highlight potential reasons for this discrepancy in two time points and explore possible limitations. Additionally, the authors should discuss the implications of these findings and provide suggestions for future research to understand the study's outcomes comprehensively.

To clear up why the relationships we observed at both time points occurred despite disordered social media use and fear of COVID-19 decreasing between time point 1 and time point 2 we explained how individual differences can occur separately to central tendencies in the data changing.

We have now noted on lines 303-308:

“Finally, the relationships discussed above were present at both time points despite disordered social media use and fear of COVID-19 decreasing between time point 1 and time point 2. However, this is because the relationship between variables does not depend on the mean score but rather the rank order of individuals. Suggesting the pattern of individual differences is present at both time points.”

Round 2

Reviewer 3 Report

The authors have responded well to all the concerns raised in the manuscript. Based on their efforts and improvements, I  recommend accepting the paper in its present form for publication.

Author Response

Dear Reviewer,

Thank you for your comments and all your advice to help us improve our manuscript.

The Editor had some suggestions and we have included our responses to those here and made some further changes to the manuscript highlighted in yellow for your information.

Kind regards,

Dr Andrew Lavender

I have some suggestions for the manuscript:

1). the outcome variables in this study are stress and depression, the term of well-being is inappropriate, you should refer to them directly and clearly in the introduction;

 We agree that we should use terminology that better reflects the aim of the study. The manuscript has been updated (with changes highlighted in yellow) to use depression and stress in places where psychological well-being was previously used.

2).  in the introduction, the Disordered Social Media Use During COVID-19 should be clearly stated clearly (for example it should be stated in the first paragraph, and the prominence of Disordered Social Media Use  During COVID-19 should also be discussed);

We thank the reviewer for this suggestion however given the length of the introduction we aimed to use the first paragraph to set the scene around the impacts of COVID. Disordered Social Media
Use is introduced on line 60 directly after COVID has been discussed.

3).  The SSO model is inappropriate, as the stress is the outcome variable, and Disordered Social Media Use is not an direct stress, and the evidences and logic of the model shoudl be further clearly stated;

 Thank you for the suggestion in terms of rethinking our guiding theory. I think in this instance the term ‘Stressor’ is being conflated with ‘Stress’. Stressors are are situations, stimuli, or events that can elicit a stress response in individuals. Where the construct of stress is the psychological experience felt by an individual (the outcome).

 4). maybe, you could test an integrated model rather than independent models.

 Thank you for the suggestion,. We think the integrated model could provide an average indirect effect coefficient and tell us about whether our effects are conditional on time. However, our conclusions in the discussion at the moment are based on the insights of the separate models as we were not specifically looking at whether the effects were conditional on time. Future work could aim to develop an integrated model, perhaps with longitudinal data designs such as the ones we discuss towards the end.
